DATA RELEASE

# Genome assembly and annotation of *Acropora pulchra* from Mo'orea French Polynesia

Trinity Conn[1,*,†], Jill Ashey[2,†], Ross Cunning[1] and Hollie M. Putnam[2]

1 Conservation Research Department, John G. Shedd Aquarium, Chicago, IL, USA
2 Department of Biological Sciences, University of Rhode Island, 120 Flagg Rd., Kingston, RI 02881, USA

## ABSTRACT

Reef-building corals are integral ecosystem engineers of tropical reefs but face threats from climate change. Investigating genetic, epigenetic, and environmental factors influencing their adaptation is critical. Genomic resources are essential for understanding coral biology and guiding conservation efforts. However, genomes of the coral genus *Acropora* are limited to highly-studied species. Here, we present the assembly and annotation of the genome and DNA methylome of *Acropora pulchra* from Mo'orea, French Polynesia. Using long-read PacBio HiFi and Illumina RNASeq, we generated the most complete *Acropora* genome to date (BUSCO completeness of 96.7% metazoan genes). The assembly size is 518 Mbp, with 174 scaffolds, and a scaffold N50 of 17 Mbp. We predicted 40,518 protein-coding genes and 16.74% of the genome in repeats. DNA methylation in the CpG context is 14.6%. This assembly of the *A. pulchra* genome and DNA methylome will support studies of coastal corals in French Polynesia, aiding conservation and comparative studies of *Acropora* and cnidarians.

**Subjects** Genetics & Genomicsm, Marine Biology, Animal Genetics

**Submitted:** 25 January 2025

\* Corresponding author. E-mail: tconn@sheddaquarium.org

† Contributed equally.

Preprint submitted at https://doi.org/10.1101/2025.03.27.645822

## DATA DESCRIPTION

Coral reefs are one of the most productive and valuable ecosystems on the planet, supporting more than 25% of marine life. The foundation of these reefs is scleractinian corals, which build calcium carbonate skeletons and harbor endosymbiotic algae (Symbiodinaceae) within their tissues. Corals around the world face mass extinction due to rising ocean temperatures from climate change. Generating high-quality coral genomes will expand available resources to investigate mechanisms through which these corals can withstand rising ocean temperatures due to climate change.

## CONTEXT

*Acropora pulchra* is a branching coral species distributed throughout the Indo-Pacific (Figure 1) within the genus *Acropora*, the most speciose genus of scleractinian corals [1]. *A. pulchra* is a hermaphroditic spawning coral species and tends to be found in fringing and back-reef habitats, showing a preference for low-wave energy environments. It can propagate asexually by fragmentation to form very large and dense thickets that provide critical habitat for reef organisms, as well as providing coastal protection. Populations of *A. pulchra*, like other Acroporids, face population loss due to rapid ocean warming [2]. The preference of *A. pulchra* for low-wave fringing and lagoon habitats, along with its close proximity to the coast and the potential for negative human impacts, makes *A. pulchra* an

**Figure 1. Image of Study Species and Geographic Distribution.**
(a) Images of *Acropora pulchra* in Mo'orea taken by Trinity Conn. (b) Map of the distribution of *Acropora pulchra* in the tropical pacific from the Coral Trait Database, edited by Joshua Madin, species ID 145 [5].

important indicator species. However, to date, most studies on the acclimatization and adaption of *Acropora spp*, and *A. pulchra* in particular, have relied on reference genomes from other *Acropora* species [3, 4]. Understanding the mechanisms through which *A. pulchra* may be resilient to climate change requires species-specific genomic resources. This study provides such resources to support further genetic and mechanistic research.

This is the first study in scleractinian corals to present the DNA methylome in tandem with a high-quality genome assembled utilizing PacBio long-read HiFi sequencing. Here, we describe the data generation, *de novo* reference genome assembly and annotation, and characterization of the DNA methylome. We release the data as a publicly available resource.

## METHODS

### Sample collection, DNA extraction, and sequencing

On 23 October 2022, sperm samples were collected from the spawning of *Acropora pulchra* and preserved in DNA/RNA shield (Zymo Research). The sperm samples were preserved by mixing 1 mL of sperm ($10^6$ sperm/mL) with 1 mL of DNA/RNA shield. Samples were exported under CITES FR2398700017-E to the University of Rhode Island and one sample was sent to the DNA Sequencing Center at Brigham Young University for extraction and long-read PacBio sequencing.

DNA was extracted by the DNA Sequencing Center at Brigham Young University using the Qiagen Genomic Tip protocol and buffers (Qiagen Cat #10223). The sample was ethanol (2×) precipitated post-column elution, put in the −20°C freezer overnight, and then centrifuged for 30 minutes at 14K rcf the following day. Ethanol was removed and the DNA

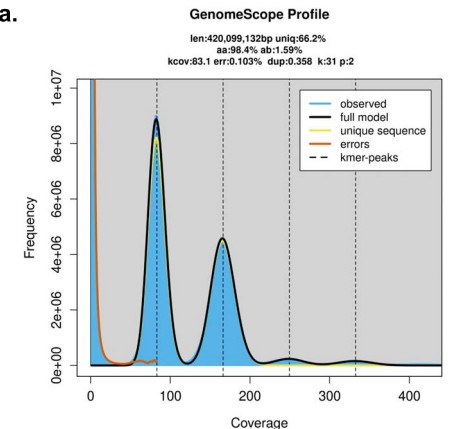
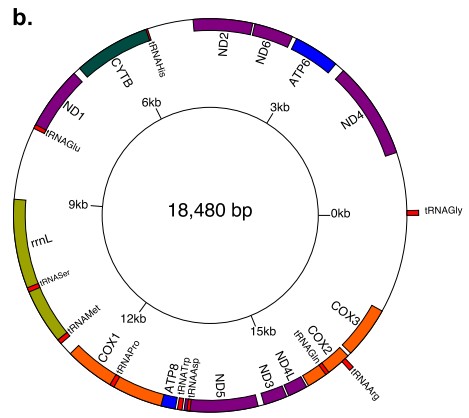

**Figure 2.** (a) K-mer multiplicity plot and estimates of genome size and heterozygosity as inferred by GenomeScope2.0. (b) Final assembly of the circularized mitochondrial genome. Genic regions in red are tRNA-protein coding genes. Genic regions in orange are COX genes. Genic regions in purple are genes related to the production of NADH dehydrogenase. Genic regions in blue are genes related to the production of ATP synthase. Genic regions in dark green produce cytochrome B. Genic regions in light green are genes involved in the generation of ribosomal RNAs. Mitochondria were circularized starting with tRNAGly.

pellets were suspended in low-EDTA Tris-EDTA buffer. The resulting DNA was cleaned prior to library preparation with the PacBio SRE Kit (PacBio Cat #102-208-300) to remove fragments under 25kb. Following extraction and cleaning, DNA was sheared to 17 kb using a Diagenode Megaruptor (Diagenode Cat #B06010003) and checked on an Agilent Femto Pulse system (Agilent Part #M5330AA) to assess size. The DNA was then cleaned and concentrated post-shearing using a 1× AMPure bead cleaning (AMPure Cat #20805800). DNA underwent library preparation using the PacBio SMRTbell prep kit 3.0 (PacBio Cat #102-141-700), following the manufacturer's instructions. The final sizing of the library was performed using the 35% v/v dilution of AMPure PB beads (AMPure Part #100-265-900). The single SMRTbell library was then sequenced using one 8M SMRT Revio Cell, and run for 29 hours on a PacBio Revio sequencer. Consensus accuracy of circular consensus sequencing processing was used to generate HiFi reads.

### Assembly cleaning, mitochondrial genome assembly

Prior to assembly, k-mers were tallied using Jellyfish v 2.2.10 (RRID:SCR_005491) [6], and distributions of k-mer size were then fed into GenomeScope v.2.0 (RRID:SCR_017014) [7] with *k* = 31 to estimate nuclear genome size and heterozygosity (Figure 2a).

To assemble the mitochondrial genome, MitoHifi v2.2 (RRID:SCR_026369) [8] was used on the raw HiFi reads generated for *A. pulchra*. The mitochondrial genome for *Acropora digitifera* (NCBI Accession BLFC01000001–BLFC01000955) was used as a seed genome for MitoHiFi with the arguments -a animal (organism-type animal) and -o 5 (invertebrate genetic code).

After mitochondrial genome assembly and removal from the raw HiFi reads, further non-coral DNA reads were identified in the raw HiFi reads using BLASTn (RRID:SCR_001598) against the following databases: NCBI common eukaryotic contaminant sequences, NCBI viral representative genome sets, NCBI prokaryotic representative genome sets, and the genomes of *Symbiodinium microadriaticum* [9] and *Durusdinium trenchii* [10].

Previous work has shown that *A. pulchra* in Mo'orea typically hosts one of these two symbiont types (Huffmeyer *et al.* in preparation). HiFi reads with a bit score >1000 were removed prior to nuclear genome assembly [11, 12].

## Nuclear genome assembly and haplotig purging

Cleaned HiFi reads were assembled into primary and alternate assemblies with Hifiasm v0.16.0 (RRID:SCR_021069) [13] with the following parameters: -primary, -s .55. Scaffolding of the primary assembly was done using nt-links v1.3.11 [14] with the following parameters: -g 100, -rounds 5, -gap_fill. Putative haplotigs were removed using HiFiasm's internal purge_dups pipeline. Statistics for primary and alternate assemblies were calculated using Quast v.5.3.0 (RRID:SCR_001228) [15].

## Structural and functional annotation

To identify and mask Transposable Elements (TEs) and repeat content, a repeat database was modeled using RepeatModeler v.2.0.6 (RRID:SCR_015027) [16]. The flag –LTRStrcut was added to detect Long Terminal Repeats (LTRs). A comprehensive repeat library was generated using this database, the protein reference library from *A. digitifera*, and the RepeatMasker database of common TEs. The clean assembly was then soft-masked using this repeat library in RepeatMasker v4.1.5 (RRID:SCR_012954) [17]. Assembly completeness of the soft-masked genome was assessed using BUSCO v.5.8.2 (RRID:SCR_015008) [18] and benchmarked against 954 Metazoan orthologs from 65 genomes using the database metazoa_odb10.

The soft-masked assembly for *A. pulchra* was annotated using funannotate v1.8.13 (RRID:SCR_023039) [19]. Raw RNAseq reads from *A. pulchra* (NCBI BioProject Accession PRJNA1201098) were used as input for *ab initio* gene predictions. Prior to gene prediction, RNAseq data were trimmed using fastp v0.19.7 (RRID:SCR_016962) [20] as described by Becker et al. [21]. To generate *ab initio* gene predictions, funannotate train was first run on the *A. pulchra* assembly with RNAseq data with the following parameters: –maxintronlength of 100000, –cpus 20, –pasa_db sqlite –memory 300G –species 'Acropora pulchra'. Funannotate train uses Trinity v.2.15.2 (RRID:SCR_013048) [22] and PASA v.2.5.3 (RRID:SCR_014656) [23] for transcript assembly prior to *ab initio* predictions. Funannotate predict was then run for *ab initio* gene models using AUGUSTUS v.3.5.0 (RRID:SCR_008417) [24] and GeneMark (RRID:SCR_011930) v.4.72 [25], and this evidence was input into Evidence Modeler v.2.1.0 (RRID:SCR_014659) [23] to generate consensus gene models. The gene models and untranslated regions predicted from the funannotate predict wrapper were refined using the previous RNAseq data and two runs of PASA with the funannotate update wrapper. Gene function and ontology were first assigned using InterProScan v/5.71-102 (RRID:SCR_005829) [26] and EggNOG v.2.1.12 (RRID:SCR_002456) [27]. The resulting outputs were used as additional input for the funannotate annotate wrapper function, which annotates gene function and ontology to protein-coding gene models using the InterProScan and EggNOG results, and a curated database to parse Pfam (RRID:SCR_004726) [28], CAZyme (RRID:SCR_012909) [29], UniProtKB (RRID:SCR_004426) [30] and GO (RRID:SCR_002811) [31] annotations.

## Detection of genome-wide methylation

Sequences in the raw PacBio bam file were aligned to the assembled, scaffolded and soft-masked genome using pbmm2 v1.16.99 (RRID:SCR_025549) [32]. The aligned bam file

**Table 1.** Comparison of assembly statistics of this study and other Acroporid assemblies assemblies. The * indicates the assembly presented in this paper.

| Statistic | *Acropora pulchra** | *A. pulchra,* Guam | *Acropora millepora* | *Acropora digitifera* | *Acropora hyacinthus* | *Acropora cervicornis* | *Acropora palmata* |
|---|---|---|---|---|---|---|---|
| Total size (Mbp) | 518.3 | 459.2 | 475.4 | 447.5 | 495.2 | 388.9 | 335.9 |
| GC content (%) | 39.1 | 39 | 39 | 39 | 39 | 39 | 39 |
| No. of scaffolds | 174 | 1340 | 854 | 955 | 153 | 159 | 351 |
| Scaffold N50 (Mbp) | 17.8 | 30.8 | 19.8 | 0.438 | 35.2 | 22.2 | 22.6 |
| Scaffold L50 | 10 | 7 | 9 | 272 | 7 | 7 | 7 |
| No. gene models | 45,061 | N/A | 42,775 | 32,106 | N/A | 39,685 | 40,233 |
| No. protein coding genes | 40,518 | N/A | 30,136 | 26,073 | N/A | 35,454 | 34,668 |
| Accession | PRJNA1162071 | GCA_965118204.1 | PRJNA633778 | BLFC01000001– BLFC01000955 | GCA_964291705.1 | GCA_964034795.1 | GCA_964030595.1 |
| Citation | This study | GCA_965118204.1 | Fuller *et al.* [37] | Shinzato *et al.* [38] | GCA_964291705.1 | GCA_964034795.1 | GCA_964030595.1 |

was used as input for pb-CpG-tools v2.3.2, which calculates methylation probabilities for each CpG site [33]. CpG sites with <5× coverage were filtered out of consideration for subsequent analyses to increase confidence in the methylation calls. CpGs were classified as being highly methylated (≥50% methylation), partially methylated (10–50% methylation), or lowly methylated (≤10% methylation), following methods previously used to describe coral DNA methylation in other species [34]. CpG locations were then intersected with flanking (1000 bp up and downstream of the genes), exonic, intronic, and intergenic regions using BEDTools v2.30.0 (RRID:SCR_006646) [35] to determine the relative distribution and quantity.

## RESULTS

### Mitochondrial genome assembly

The mitochondrial genome was successfully assembled and circularized using MitoHifi v3.2.2. The final assembled *A. pulchra* mitogenome is 18,480 bp in length with 15 protein-coding genes. The *A. pulchra* mitogenome contains nine transfer RNA (tRNA) coding genes (Figure 2b).

### Genome assembly

PacBio sequencing of high molecular weight DNA from *A. pulchra* sperm produced 5,898,386 total HiFi reads with a mean read length of 13,425 bp and a total length of 79,183,709,778 bp. BLASTn identified 1,922 reads as non-coral in origin (algal eukaryotic, viral, prokaryotic, and mitochondrial) with a bit score >1000 (0.03% of total reads). After contaminants were removed, 5,896,464 HiFi reads remained with a mean read length of 13,426 bp and a total read length of 79,168,047,656. The remaining HiFi reads were used in the genome assembly.

The primary assembly is 518,528,298 bp in length, consisting of 188 contigs and a GC content of 39.05%. The N50 of the primary assembly is 16.26 Mbp and L50 is 11. The total length of the final scaffolded assembly is 518,313,916 bp, consisting of 174 total scaffolds and a GC content of 39.1%. The largest scaffold is 45,111,900 bp, and this genome is highly contiguous with an N50 of 17,861,421, L50 of 10 (Table 1).

### Structural and functional annotation

Annotation of the repeat content of the genome found 16.74% of the genome to be repetitive (Table 2), with the majority of repetitive regions unclassified (11.24%) (Table 2). This is

**Table 2.** Summary of Repeat Content Statistics as determined by RepeatMasker.

| Element | Number of Elements | Length (bp) | Percentage of Genome |
|---|---|---|---|
| Retroelements: | 53,931 | 15,969,347 | 3.08% |
| SINEs | 12,165 | 1,556,022 | 0.30% |
| Penelope | 1,664 | 463,158 | 0.09% |
| LINEs | 37,300 | 12,835,432 | 2.48% |
| **CRE/SLACS** | 550 | 214,445 | 0.04% |
| **L2/CR1/Rex** | 18,647 | 5,737,962 | 1.11% |
| **R2/R4/NeSL** | 869 | 368,129 | 0.07% |
| **RTE/Bov-B** | 399 | 97,188 | 0.02% |
| **L1/CIN4** | 441 | 341,464 | 0.07% |
| LTR Elements | 4,466 | 1,577,793 | 0.30% |
| **BEL/Pao** | 420 | 169,263 | 0.03% |
| **Ty1/Copia** | 592 | 156,898 | 0.03% |
| **Gypsy/DIRS1** | 2,474 | 859,774 | 0.17% |
| DNA transposons: | 18,539 | 4,870,082 | 0.94% |
| **hobo-Activator** | 857 | 158,746 | 0.03% |
| **Tc1-IS630-Pogo** | 5,587 | 1,207,096 | 0.23% |
| **Tourist/Harbinge** | 3,339 | 865,433 | 0.17% |
| Rolling-circles | 865 | 182,230 | 0.04% |
| Unclassified | 288,287 | 58,251,167 | 11.24% |
| Total Interspersed repeats | N/A | 79,553,654 | 15.35% |
| Small RNA | 18,443 | 2,403,369 | 0.46% |
| Satellites | 888 | 97,262 | 0.02% |
| Simples repeats | 109,119 | 4,909,261 | 0.95% |
| Low complexity | 15,646 | 742,738 | 0.14% |

comparable to other Acroporid species [36–38]. The initial run of funannotate predict resulted in the annotation of 44,371 genes, with an average gene length of 4,848.65 bp, and an average exon length of 212.27 bp. After running funannotate update, annotations were updated to a final total of 45,061 gene models, with an average gene length of 5,443.66 bp and an average exon length of 277.68 bp. Running funannotate annotate resulted in 22,115 interproscan annotations, 23,804 EggNOG annotations, 21,052 Pfam domains and 575 CAZyme annotations.

## Genome-wide methylation

There were 15,939,114 CpG sites identified in the genome, with 15,928,397 CpG sites remaining after sites with <5× coverage were removed. Using a cutoff of >50% methylation to define methylated CpGs, 14.6% of CpGs were methylated, 0.2% were partially methylated (10–50% methylation), and 82.3% were lowly methylated (<10% methylation).

The percentage of CpG sites was evaluated in genomic features, specifically intergenic regions, exons, introns, downstream flanks, and upstream flanks. The majority of CpG sites were found in gene body and flanking regions (61.7%), while 38.3% were identified in intergenic regions (Figure 4a).

The analysis of CpG site methylation revealed varying levels of methylation across genomic features. Highly methylated CpGs (>50%) were most prevalent in introns (17.1%), followed by intergenic regions (13.7%), upstream flanking regions (13.0%), downstream flanking regions (12.9%) and exons (12.7%; Figure 4b).

Partially methylated CpGs (10–50%) were comparatively rare, with proportions ranging from 2.3% in exons to 3.9% in intergenic regions. In contrast, the majority of CpGs were unmethylated (<10%), accounting for 82.3%–85.0% of sites across all features, with the

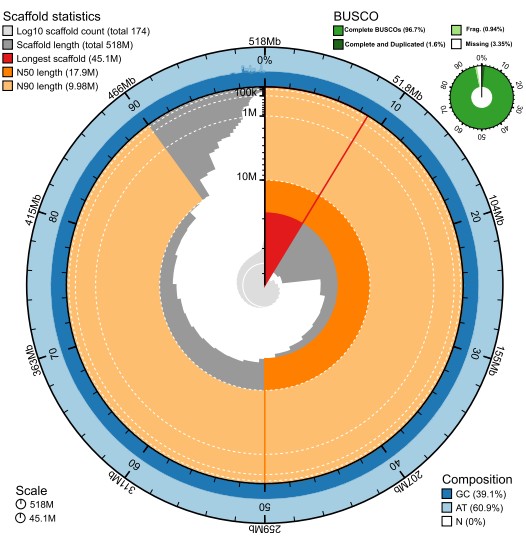

**Figure 3.   Snail plot of Statistics Across the Final Masked Genome.**
Nucleotide composition presented in dark (GC content) and light (AT content) blue. GC content was consistent across the entire genome at approximately 39.1%. Scaffold length statistics are presented in orange, red, and grey.

lowest proportion in introns (80.5%; Figure 4b). The methylation probability across the first 100,000 bp of the ten largest scaffolds is shown in Figure 4c.

## DATA VALIDATION AND QUALITY CONTROL

A BUSCO analysis of the final assembled genome against a benchmark of 954 conserved metazoan orthologs found 95.1% complete and single-copy BUSCOs, 1.6% complete and duplicated BUSCOs, 0.9% fragmented, and 2.4% missing (Figure 3), making the *Acropora pulchra* genome one of the most complete Acroporid assembly to date (Figure 5).

## REUSE POTENTIAL

This *Acropora pulchra* genome assembly represents the most complete Acroporid genome assembly to date, as well as one of the most contiguous Acroporid genomes, and will serve as a resource to the coral and wider scientific community. A complete and contiguous genome will improve the accuracy of alignment, identification, and assignment of gene function, and improve the general understanding of the biology of this species. Furthermore, this quantification of the genome-wide methylation facilitates the capacity to study the epigenetics of non-model organisms and, specifically, future analyses on methylation in *A. pulchra*.

## DATA AVAILABILITY

Raw reads, metadata, and genome assembly are available on NCBI at BioProject PRJNA1162071 and BioSample SAMN43800006 (version 1.1), and NCBI Genome Accession GCA_044231415.1. RNA-seq data is available under NCBI BioProject Accession PRJNA1201098. All related files and scripts used to assemble and annotate the genome can be found on the Open Science Framework at [39].

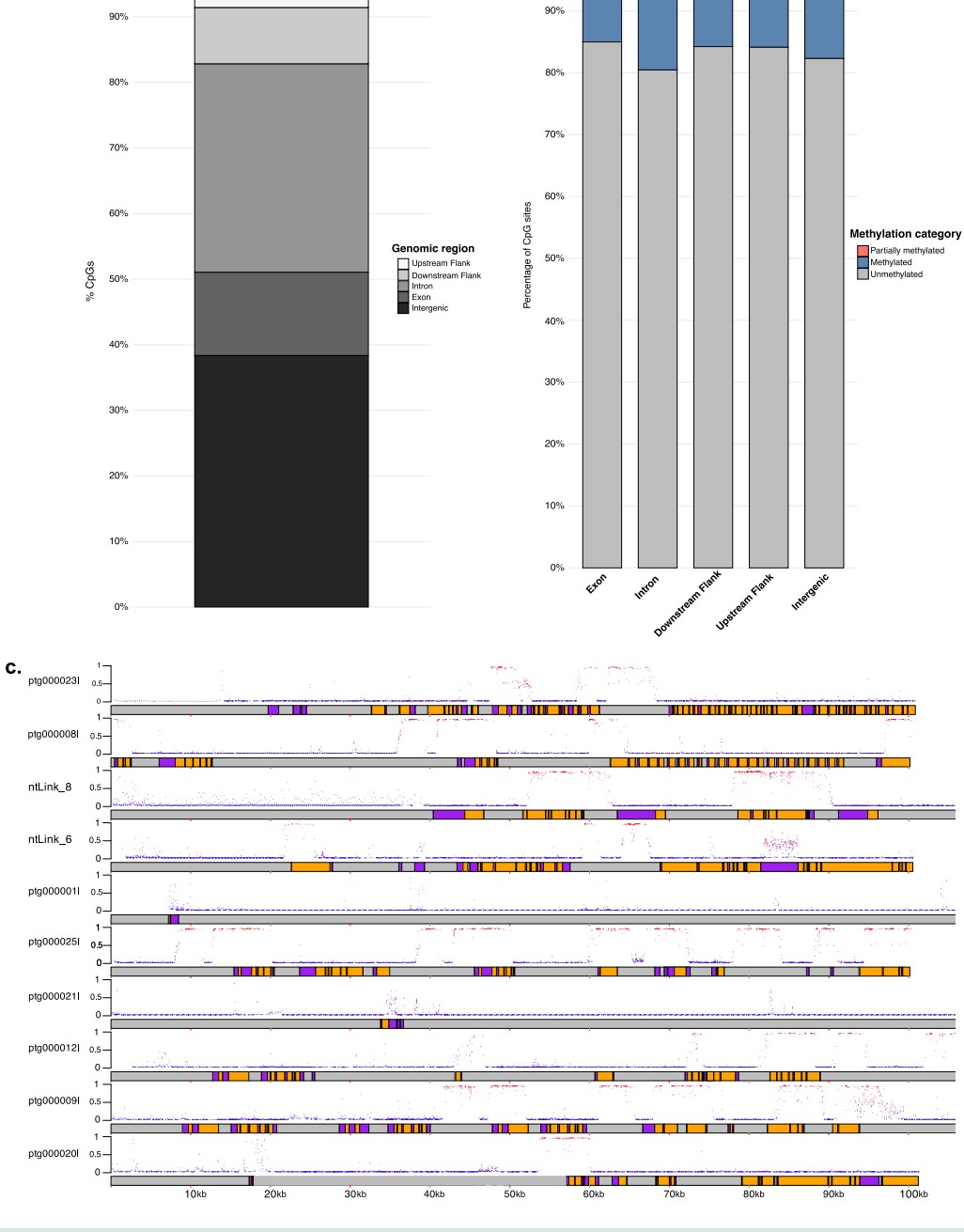

**Figure 4.** (a) Distribution of CpG sites across genomic features. (b) Percentage of CpGs methylated, partially methylated, and unmethylated in exons, introns, and intergenic regions. (c) Distribution of CpGs and their methylation probability across the first 100,000 bp of the ten longest scaffolds in the *Acropora pulchra* genome. Purple sections are exons, orange sections are introns, and gray sections are intergenic regions. The scaffold name is to the left of each scaffold. Each point represents a CpG location. The color and height (0 to 1) of the point on the y-axis represent the probability that the specific CpG site is methylated. Red indicates a higher probability of methylation, and blue indicates a lower probability of methylation.

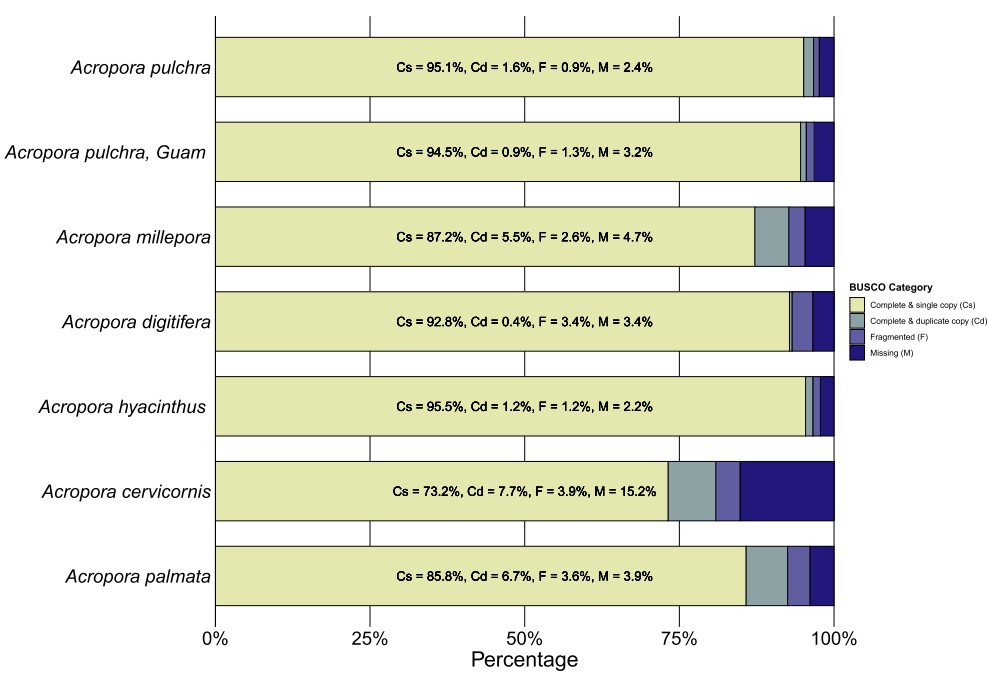

**Figure 5.** BUSCO Completeness of 954 conserved metazoan orthologs across *Acropora* Genome Assemblies. See Table 1 for the citations of each genome. The genome of *Acropora pulchra* presented in this paper represents the most complete Acroporid genome to date, with 95.1% complete and single-copy BUSCOs, 1.6% complete and duplicated BUSCOs, 0.9% fragmented, and 2.4% missing.

## LIST OF ABBREVIATIONS

LTR, Long Terminal Repeat; TE: transposable element; tRNA: transfer RNA.

## DECLARATIONS

### Ethical approval

Samples were exported under CITES FR2398700017-E.

### Consent for publication

Not applicable.

### Competing interests

The authors declare that they have no competing interests.

### Authors' contributions

TC: Formal Analysis, Writing (Original Draft), Visualization, Methodology, Investigation; JA: Formal Analysis, Writing (Original Draft), Data Curation, Visualization, Methodology, Investigation; RC: Funding Acquisition, Conceptualization, Writing (Edits); HMP: Funding Acquisition, Conceptualization, Data Curation, Writing (Edits).

### Funding

This work was supported by NSF grant EF-1921425 to RC, and NSF grant EF-1921465 to HMP. JA was supported by the NSF Graduate Research Fellowship Program. This work represents

a contribution of the Mo'orea Coral Reef LTER Site and was supported by resources from NSF-OCE 2224354 to the Mo'orea Coral Reef LTER, as well as a generous gift from the Gordon and Betty Moore Foundation.

## Acknowledgements

We would like to thank the UC Berkeley Gump South Pacific Research Station Station for logistical support and facility space. We thank Ariana Huffmyer, Pierrick Harnay, and Danielle Becker for field assistance, as well as Zoe Dellaert and Danielle Becker for laboratory assistance. We also thank the University of Rhode Island High Performance Computing for its logistical support. Research was completed under permits issued by the French Polynesian Government (Délégation à la Recherche) and the Haut-commissariat de la République en Polynésie Française (DTRT) (Protocole d'Accueil 2005–2023) and CITES FR2398700017-E. As guests, we recognize and give thanks for the land and water resources of Polynesia, in particular Mo'orea, and to the traditional owners of the land, both past and present. Māuruuru roa. With respect to the spelling of Tahitian words, we endeavored to follow the Te Fare Vāna'a transcription system that is adhered to by a large segment of the Tahitian community, but also recognize other community members follow the Raapoto transcription system where the island name of Moorea is, for example, spelled without the 'eta (i.e., Moorea).

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
