## [Reviewer Report]

Indicate in the comments box below whether you are happy with the changes made or if the manuscript is unacceptable.Comments on revised manuscriptHi there The authors have addressed all my comments and queries, and included nearly all recommendations. Thank you !
A few quick notes to fix before publication - "The input created Funannotate train uses Trinity v.2.15.2 [22] and PASA v.2.5.3 [23] for transcript assembly prior to ab initio predictions". This sentence reads weird, reword before publishing. I think maybe just remove "created Funannotate train" and then it reads correctly. Or "Funnannotate trains uses .....". - "PFAM v.37.0 [28], CAZyme [29], UniProtKB v[30] and GO [31]." Missing a few version numbers, and UniProt just has a v. - "The mitochondrial genome was successfully assembled and circularized using MitoHifi v3.2.2 The final assembled A. pulchra mitogenome is". Just missing a period i think before "The final assembly". Great job and a very useful resource for the coral community !!Indicate in the comments box below whether you are happy with the changes made or if the manuscript is unacceptable.Comments on revised manuscriptHi there The authors have addressed all my comments and queries, and included nearly all recommendations. Thank you !
A few quick notes to fix before publication - "The input created Funannotate train uses Trinity v.2.15.2 [22] and PASA v.2.5.3 [23] for transcript assembly prior to ab initio predictions". This sentence reads weird, reword before publishing. I think maybe just remove "created Funannotate train" and then it reads correctly. Or "Funnannotate trains uses .....". - "PFAM v.37.0 [28], CAZyme [29], UniProtKB v[30] and GO [31]." Missing a few version numbers, and UniProt just has a v. - "The mitochondrial genome was successfully assembled and circularized using MitoHifi v3.2.2 The final assembled A. pulchra mitogenome is". Just missing a period i think before "The final assembly". Great job and a very useful resource for the coral community !!

---

## [Editor Report]

Editor’s AssessmentAcropora pulchra is a species small polyped stony corals in the family Acroporidae from the the Indo-Pacific. This Data Release is the first study in stony corals to present the DNA methylome in tandem with a high-quality genome assembled utilizing PacBio long-read HiFi sequencing. Sequencing an A. pulchra specimen from Mo’orea, French Polynesia. From this single molecule sequencing data DNA methylation data was also called and quantified, and additional short-read Illumina RNASeq data was used for gene annotation. This producing an assembly size is 518 Mbp, with 174 scaffolds, and a scaffold N50 of 17 Mbp, and 40,518 protein-coding genes called. Peer review requested some improved benchmarking, and it is impressive to see from the results that the genome assembly represents the most complete and contiguous stony coral genome assembly to date. As an important indicator species and this data will hopefully serve as a resource to the coral and wider scientific community. Further quantification of the genome-wide methylation is needed aid the study epigenetics of non-model organisms, and specifically future analyses on methylation in coral.Editor’s AssessmentAcropora pulchra is a species small polyped stony corals in the family Acroporidae from the the Indo-Pacific. This Data Release is the first study in stony corals to present the DNA methylome in tandem with a high-quality genome assembled utilizing PacBio long-read HiFi sequencing. Sequencing an A. pulchra specimen from Mo’orea, French Polynesia. From this single molecule sequencing data DNA methylation data was also called and quantified, and additional short-read Illumina RNASeq data was used for gene annotation. This producing an assembly size is 518 Mbp, with 174 scaffolds, and a scaffold N50 of 17 Mbp, and 40,518 protein-coding genes called. Peer review requested some improved benchmarking, and it is impressive to see from the results that the genome assembly represents the most complete and contiguous stony coral genome assembly to date. As an important indicator species and this data will hopefully serve as a resource to the coral and wider scientific community. Further quantification of the genome-wide methylation is needed aid the study epigenetics of non-model organisms, and specifically future analyses on methylation in coral.

---

## [Reviewer Report]

Reviewer name and names of any other individual's who aided in reviewer Yanshuo LiangDo you understand and agree to our policy of having open and named reviews, and having your review included with the published papers. (If no, please inform the editor that you cannot review this manuscript.)YesIs the language of sufficient quality?YesPlease add additional comments on language quality to clarify if needed
Are all data available and do they match the descriptions in the paper? YesAdditional CommentsAre the data and metadata consistent with relevant minimum information or reporting standards? See GigaDB checklists for examples <a href="http://gigadb.org/site/guide" target="_blank">http://gigadb.org/site/guide</a>YesAdditional CommentsIs the data acquisition clear, complete and methodologically sound?YesAdditional CommentsIs there sufficient detail in the methods and data-processing steps to allow reproduction?YesAdditional CommentsIs there sufficient data validation and statistical analyses of data quality? YesAdditional CommentsIs the validation suitable for this type of data?YesAdditional CommentsIs there sufficient information for others to reuse this dataset or integrate it with other data?YesAdditional CommentsAny Additional Overall Comments to the AuthorThe manuscript by Conn et al. detail the high-quality genome assembly of Acropora pulchra, a Acropora of ecological and evolutionary significance, and also analyzes its genome-wide DNA methylation characteristics. These data complement the genetic resources of the Acropora genome. This manuscript is well written and represents a valuable contribution to the field. I have some comments below for the authors to address but look forward to seeing this research published. Q1: In the first sentence of the second paragraph of the Context: This is the first study to utilize PacBio long-read HiFi sequencing to generate a high quality genome with high BUSCO completeness, in tandem with its DNA methylome for scleractinian corals. Language such as "new", "first", "unprecedented", etc, should be avoided because it often leads to unproductive controversy. As far as I know, the genome you assembled is not the first stony coral to be sequenced using PacBio long-read HiFi sequencing. Back in 2024, He et al. assembled Pocillopora verrucosa (Scleractinia) to the chromosome level using PacBio HiFi long-read sequencing and Hi-C technology. Here I would suggest please rephrase. Reference： He CP, Han TY, Huang WL, et al. Deciphering omics atlases to aid stony corals in response to global change, 11 March 2024, PREPRINT (Version 1) available at Research Square [https://doi.org/10.21203/rs.3.rs-4037544/v1]. Q2: In this sentence: “On 23 October 2022, sperm samples were collected from the spawning of A.pulchra and preserved in Zymo DNA/RNA shield.” Please “A.pulchra” to “A. pulchra”. Q3: Please change all “k-mer” into “k-mer” in the manuscript. Q4: Please change “Long-Tandem Repeats” to “Long Terminal Repeats” Q5: In this sentence: “Funannotate train uses Trinity [18] and PASA [19] for ab initio predictions. Funannotate predict was then run to assign gene models using AUGUSTUS [20], GeneMark [21], and Evidence Modeler [19] to estimate final gene models.” Please write versions of these software. Q6: [20] Later references do not correspond well in the manuscript, please check!RecommendationMinor Revision

---

## [Reviewer Report]

Reviewer name and names of any other individual's who aided in reviewer Jason SelwynDo you understand and agree to our policy of having open and named reviews, and having your review included with the published papers. (If no, please inform the editor that you cannot review this manuscript.)YesIs the language of sufficient quality?YesPlease add additional comments on language quality to clarify if needed
There are some minor grammatical issues throughout that warrent a closer reading to correct. E.g. Abstract: "...urgency to identify how genetic, epigenetic, and environmental...", "...management and and conservation...". Context: "...we aim to provide..." etc.Are all data available and do they match the descriptions in the paper? YesAdditional CommentsThe link to the OSF repository in the PDF did not work. However, the link to the OSF repository from the github did work.Are the data and metadata consistent with relevant minimum information or reporting standards? See GigaDB checklists for examples <a href="http://gigadb.org/site/guide" target="_blank">http://gigadb.org/site/guide</a>YesAdditional CommentsIs the data acquisition clear, complete and methodologically sound?NoAdditional CommentsIt isn't mentioned in the manuscript where the RNAseq data used to annotate the genome is from, nor any quality filtering steps that may have been applied to the RNA data prior to its use for annotation.Is there sufficient detail in the methods and data-processing steps to allow reproduction?YesAdditional CommentsExcluding the above comment about the RNA data.Is there sufficient data validation and statistical analyses of data quality? YesAdditional CommentsIs the validation suitable for this type of data?YesAdditional CommentsIs there sufficient information for others to reuse this dataset or integrate it with other data?YesAdditional CommentsAny Additional Overall Comments to the AuthorThis is a well assembled, and annotated genome that will contribute to the growing database of Acropora genomes. The manuscript could do with a simple pass to identify and correct some relatively minor grammatical issues and inconsistencies (Table 1 includes a thousands comma separator in some instances and not others) and needs to include details about the source of the RNA data used to train the ab initio gene predictors. There also appears to be a problem with the citation numbering after 20.RecommendationMinor Revision

---

## [Reviewer Report]

Upload additional filesDRR-202501-02-R01/stage_files/DRR-202501-02/Review MS/connetal2025_reviewcomments.pdfReviewer name and names of any other individual's who aided in reviewer Benjamin YoungDo you understand and agree to our policy of having open and named reviews, and having your review included with the published papers. (If no, please inform the editor that you cannot review this manuscript.)YesIs the language of sufficient quality?YesPlease add additional comments on language quality to clarify if needed
Are all data available and do they match the descriptions in the paper? YesAdditional CommentsRaw reads, metadata, and genome assembly are publicly available and have a NCBI project number in which they are all linked.Are the data and metadata consistent with relevant minimum information or reporting standards? See GigaDB checklists for examples <a href="http://gigadb.org/site/guide" target="_blank">http://gigadb.org/site/guide</a>YesAdditional CommentsIs the data acquisition clear, complete and methodologically sound?YesAdditional CommentsCollection of sperm samples, HMW DNA extraction, and SMRT Bell Library prep are written clearly. I have asked for a few clarifications on wording in this section in the attached edited pdf document.Is there sufficient detail in the methods and data-processing steps to allow reproduction?YesAdditional CommentsI think the pipeline used for de-novo genome generation (including raw read cleaning and assembly), repeat masking, and gene prediction and annotation is of high quality and best practices. With the inclusion of the GitHub and all analyses scripts, it is possible to reproduce the assembly generated.Is there sufficient data validation and statistical analyses of data quality? YesAdditional CommentsThis is not super relevant for a genome assembly paper so I have no additional comments here.Is the validation suitable for this type of data?YesAdditional CommentsThe authors use tools such as GenomeScope2 and BUSCO for validation of their data. It would be nice to see the tool they used to identify N50 and L50 (maybe Quast) included in the methods. Additionally, I would like to see a Merqury analysis of the HifiAsm primary and alternate assemblies to show that duplicate purging was successful.Is there sufficient information for others to reuse this dataset or integrate it with other data?YesAdditional CommentsAny Additional Overall Comments to the AuthorI would first like to commend the authors for a well assembled genome resource for a coral species that will be greatly beneficial to the wider coral and scientific community. I have provided a PDF with comments throughout for the authors to address. The majority of these are easy fixes, including things such as sentence structure, inconsistent capitalisation of subheadings, additional references for methods, clarification of statements, and other suggestions. I do have a few larger requests for this to be published, and these are the reasons for selecting the major revision option as there may need to be figure updates, and quick additional analyses to be run. 1. Can you please correct the verbiage around BUSCO analysis throughout the manuscript. It is often stated "BUSCO completeness of xx%". BUSCO doesn't directly measure completeness, rather completeness of single copy orthologs against a specific database. I have left comments throughout on potential rewording for these instances. Please also specify the exact database you used (i.e. odb10_metazoa). Finally, can you please be more specific when stating BUSCO results, specifically when you use 96.9% this is single copy and duplicated complete BUSCOS. I have left comments in the pdf again for this. 2. In the results for Genome Assembly section can you please include results (i.e. length, N50, L50, number contigs/scaffolds) for the primary assembly and the scaffolded assembly. 3. I think it would be not much work and provide additional information to show successful duplicate purging to run a Merqury analysis on the primary and alternative assemblies from HiFiAsm. 4. Can you include some additional information in the "Structural and Functional Annotation section". Specifically, can you provide information on the results from the funannoatate predict step, and then how funannotate update improved this (if at all). 5. Please double check the methods section for funannotate. From reading the funannoatate documentation I think there may be some confusion on what each step (train, predict, update, annotate) is doing. I have provided comments in the pdf to help clarify, and have also linked the funnannotate documentation. 6. On NCBI I see that an additional Acropora pulchra genome has just been made available (29th Jan 2025), with this to the chromosome level (https://www.ncbi.nlm.nih.gov/datasets/genome/GCA_965118205.1/). I think it would be prudent to include this assemblies statistics in your Table 1, and also run a BUSCO analysis on this other assembly to compare with your one. While they got to chromosome level, you do have markedly less contigs. I do not think this is necessary for this manuscript, but future work you could look to use their chromosome assembly to get your scaffolded assembly to chromosome level. Again, I want to say this is a wonderful resource for the coral and wider scientific community, and the pipeline for de-novo assembly and annotation is best practices in my opinion.RecommendationMajor Revision